# A Study on the Adsorption Mechanism and Compactness of the TFS Coating Interfacial Layer

**Yafei Xie** [1,2,*], **Tong Zhao** [1,*] and **Kai Liu** [1]

---

1 School of Mechanical and Precision Instrument Engineering, Xi'an University of Technology, Xi'an 710048, China; kliu@mail.xaut.edu.cn
2 School of Intelligent Engineering, Jiangsu Vocational College of Information Technology, Wuxi 214153, China
* Correspondence: xieyafei324@163.com (Y.X.); tongzhao@xaut.edu.cn (T.Z.)

**Abstract:** Chrome-plated plates, also known as tin-free plates (TFS), are the latest substrates for coating plates. The coating plate cannot be separated from the TFS during the stamping and extension process, and the interface layer of the TFS coating plate cannot produce pores to ensure good corrosion resistance and the appearance of the metal packaging cans. This requires the TFS coating plate interfacial layer to have good adsorption and compactness. In this paper, the molecular simulation model of the interfacial layer interaction of the TFS coating plate was established by using molecular mechanics simulation, Monte Carlo simulation, and molecular dynamics simulation, and the influential rules of chromium oxide crystalline structure, coating functional group type, and coating pressure on the adsorption and compactness of interfacial layer were analyzed and verified by experiments. The results show that the adsorption is stronger when the surface of the TFS is a chromium oxide (110) crystalline surface and contains hydroxide ions. The adsorption of polyester polyurethane coating and polyether polyurethane coating for and the adsorption of polyester polyurethane coating functional groups is stronger than polyether functional groups, and the adsorption of other functional groups is ranked by the same method. The interfacial layer compactness increases with an increase in coating pressure. For this experimental sample, the value of the film pressure sensor is 18,940 g when meeting the requirements of adsorption and compactness of the interfacial layer of the TFS coating plate, which can be extended for other coating plates.

**Keywords:** adsorption mechanism; compactness; TFS; coating plate

## 1. Introduction

The dominant metal substrate for coating plates is tin-plated plates. Due to the depletion of tin resources, many scholars have started to study tin-free steel sheets (TFS). TFS substrates have been used for more than 30 years [1]. The use of TFS substrates requires theoretical studies. P. E. Pierce et al. studied the rheological principles of coatings in 1966 and investigated the thixotropic behavior of coatings, which can also provide some theoretical reference for the study of coating plates [2]. In 1986, Atsuo Tanaka et al. analyzed the effect of PET-BO residues in PET films on the adsorption of interfacial layers formed by TFS and films [3]. In 1992, G. M. Ingo et al. analyzed the adsorption mechanism of the hydrate as well as the oxide of the tin-plated plate by electrochemical and XPS methods [4]. The mechanism of adsorption of the formed interfacial layer was complicated by the experimental protocol and the experimental results could only be reproduced for their samples and were not easily reproducible. The advances in computer science over the years have provided the basic conditions for analyzing the mechanical properties of microscopic interfacial layers using molecular dynamics methods. In 2009, Y. Ye et al. [5] studied film adsorption by a laser scratching method, and the adsorption force magnitude varied with the laser power in a qualitative study. In 2018, Christopher Melvin et al. analyzed the effect of different surface treatments on the adsorption force of the interfacial layer

formed by the TFS and the film, and the surface of the sample was observed by electron microscopy [6]. In 2020, Jiyang Liu et al. [7] analyzed the adsorption energy of pet film molecules with chromium oxide using molecular dynamics methods, which are of some reference, although the polymer is solid. In 2021, J. Whiteside et al. [8] investigated the effect of uniaxial deformation on the surface morphology and corrosion properties of TFS-coated sheets using electrochemical methods. In 2022, Manoj Prabhakar et al. [9] investigated the cathodic diffusion of electrolytes on two layers of chromium coatings electrodeposited by trivalent chromium electrolytes on steel with microscopic surface defects using electron microscopy on both micro and macro scales. The above-mentioned methods used to study the adsorption of the interfacial layer of coatings cannot be directly characterized and realized in a complex way, and the study of the denseness of the interfacial layer is only limited to microscopic observation and weighing to measure the density. Advances in computer science have provided the basic conditions for analyzing the adsorption and compactness of the interfacial layer of TFS coating sheets using molecular simulation methods.

There are three methods used to study the mechanical properties of materials: the relationship between the structure and properties of materials based on quantum mechanics or density generalized theory using the method of first-nature principle calculation; the behavior of atoms and molecules based on the physical model of the atoms and molecules that make up the materials using the method of computer simulation; and data accumulated from previous experiments using statistical methods to summarize the composition, structure, and properties of the materials The statistical model of the relationship between the concepts of quantitative theory cannot directly correspond to mechanical properties and statistical methods cannot reveal the essential connections, so molecular simulation methods are chosen. Molecular simulations include molecular dynamics simulations and Monte Carlo simulations [10], depending on whether the difference between these two methods is related to continuous time [11]. We need to analyze the molecular structure that changes in continuous time, so the molecular dynamics method was chosen as the main method, and the Monte Carlo simulation method will be used as an alternative method depending on the actual situation. The molecular simulation method can analyze the molecular evolution processes in continuous time at different pressures by choosing a COMPASS III force field and NPT system synthesis.

As shown in Figure 1, the interfacial layer composed of coating and TFS is simplified and represented as three parts. The coating part consists of a polymer. The thickness of the vacuum layer is 0.5 nm. The crystalline cell has chromium oxide atoms and other atoms. The initial conditions for the molecular dynamics analysis are a temperature of 298 K and a pressure of $1.0 \times 10^{-4}$ GPa. The factors influencing the adsorption of the interfacial layer are the crystalline surface, functional groups, and hydroxide ions; the factors influencing the compactness of the interfacial layer are the morphology and density of the polymer. The purpose of this study is to find out the quantitative or qualitative relationship between the macroscopic pressure and the adsorption and compactness of the microscopic interfacial layer.

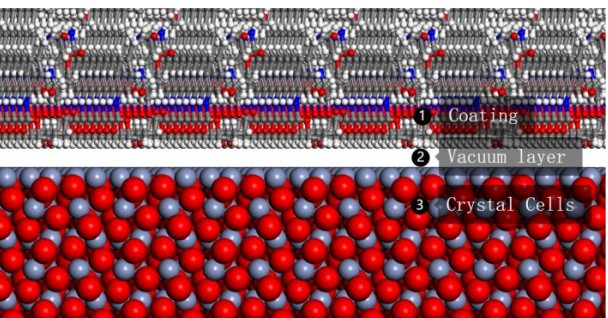

**Figure 1.** The TFS coating interfacial layer. (The gray balls are chromium atoms and the red balls are oxygen atoms).

## 2. Materials and Methods

Both the adsorption and compactness of the coating are related to the nature of the interfacial layer, which consists of microscopic molecules, ions, functional groups, and polymers. The mechanism of the adsorption and compactness needs to be dissected at the microscopic level but needs to be analyzed in relation to the macroscopic correlation of temperature, pressure, and time. Since the roll coating process operates at an ambient temperature of 25 °C all year round, we set the temperature to 25 °C and only need to adjust the process parameters at different pressure conditions. The adsorption force is characterized by the adsorption energy and the flow of the solution for the adsorption energy is shown in Figure 2.

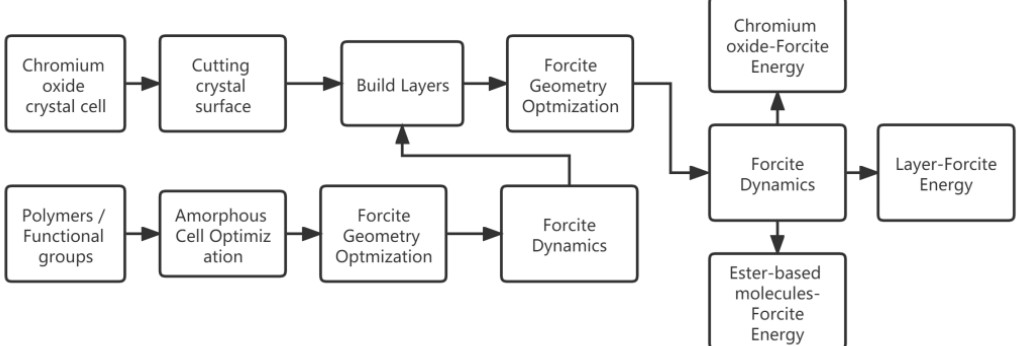

**Figure 2.** Molecular dynamics simulation flowchart.

Before the polymer is simulated, it is necessary to calculate its shape closest to the actual state using amorphous cell optimization [12], which uses the atomic structure optimization method, which satisfies the force field conditions used with the minimum distance of the atoms; its boundary condition is the density. Equation (1) is a probability calculation with corrections for the rotational state of the *i*-bond.

$$\omega_i = -\frac{E_i}{k_B T} \tag{1}$$

where $\omega_i$ is Boltzmann weight; $E_i$ is the sum of the non-bonding energy of all adjacent two atoms and the potential energy of all atoms; $k_B$ is the Boltzmann constant; and $T$ is the absolute temperature.

$$P_i = \frac{\omega_i}{\sum_{j=1}^{M} \omega_i} \tag{2}$$

where $P_i$ is the ratio of individual Boltzmann weights divided by the sum of all 1 to M Boltzmann weights:

$$q_{i-1,i}(\phi',\phi) = \frac{q_{i-1}(\phi',\phi)exp\left[\frac{-\Delta U_i}{RT}\right]}{\sum\{\phi_i\}q_{i-1}(\phi',\phi)exp\left[\frac{-\Delta U_i}{RT}\right]}, \tag{3}$$

where $q_{i-1,i}(\phi',\phi)$ denotes the conditional probability of finding a bond $i$ in state $\phi$ when the bond $i-1$ in state $\phi'$ has been determined; $\Delta U_i$ denotes the increase in non-bonding energy of the atom from state $i$ to $i+1$ based on the rotational isomeric state (RIS) model, and $p$ is the probability of a candidate state deduced using the RIS model.

$$q_{i-1,i}(\phi',\phi) = \frac{p_{i-1,i}(\phi',\phi)}{p_{i-1}(\phi')} \tag{4}$$

$$q^* = \frac{exp\left[\frac{-\Delta U_i}{RT}\right]}{\sum\{\phi_i\}exp\left[\frac{-\Delta U_i}{RT}\right]} \tag{5}$$

Equation (1) is applicable to polymer molecules with more than four primary bonds, while Equation (3) is suitable for polymer molecular formulae with more than two bonds. The polymer molecular formula model with minimum atomic distance and optimal bulk structure is obtained under the boundary conditions of the selected force field and specified density.

According to the Forcite method derived from classical mechanics [13], the molecular simulation model can be geometrically optimized, energy optimized, and kinetically simulated to finally obtain the adsorption energy. The equations of motion of classical mechanics can be expressed in several ways, and the Forcite method chooses the Hamiltonian equation to describe the energy conversion of the system.

$$E = H(q, p, b) = T(p) + U(q) + J(b) \tag{6}$$

where $E$ is the total energy, $p$ is the position, and $q$ is the momentum; $b$ is the bond distance; $T$ is the potential energy; $U$ is the kinetic energy; and $J$ is the non-bond energy. Equation (6) corresponds to the energy conversion process, as reflected in the software simulation in Figure 3. When the total energy is stable, the morphology of the polymer molecule is closest to the actual state.

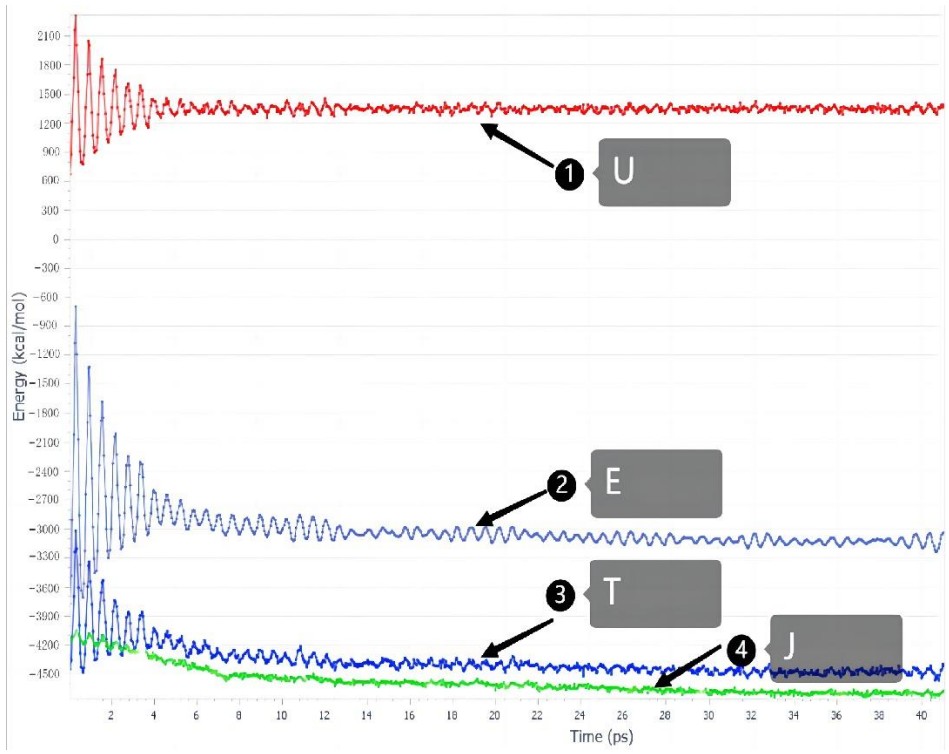

**Figure 3.** Energy variation in the interfacial layer dynamics calculation.

According to the optimization of the polyester-type polyurethane molecule, the molecular formula changes, as shown in Figure 4. The molecular formula becomes more compact to reach the state of minimum energy. Amorphous cell optimization to Forcite method optimization results inchanges occurring that are relatively small, but the measurement of the same position of the two atomic distance changed, indicating that optimization occurs.

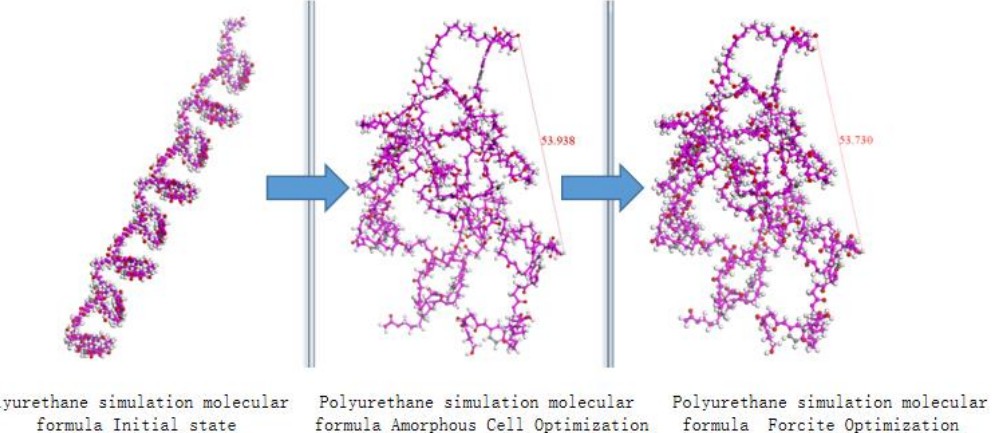

Polyurethane simulation molecular formula Initial state · Polyurethane simulation molecular formula Amorphous Cell Optimization · Polyurethane simulation molecular formula Forcite Optimization

**Figure 4.** Polyester molecular optimization process demonstration. (Different colored balls represent different atoms).

## 3. Discussion of Interfacial Layer Adsorption Force

### 3.1. The Effect of Crystal Surface on the Adsorption Force

Figure 5 shows the composition structure of the TFS in a manufacturer's product specification. The substrate after removing the oil film is chromium oxide atoms, and the chromium oxide atomic layer shows different crystal surfaces. Xiaolu Pang et al. [14] finally obtained $Cr_2O_3$ (012) by different production methods, and the article can show one point that $Cr_2O_3$ with different lattice indices can be obtained with different production processes; so, we can analyze in the microscopic world which crystal surface of $Cr_2O_3$ has the largest adsorption force with the coating molecules.

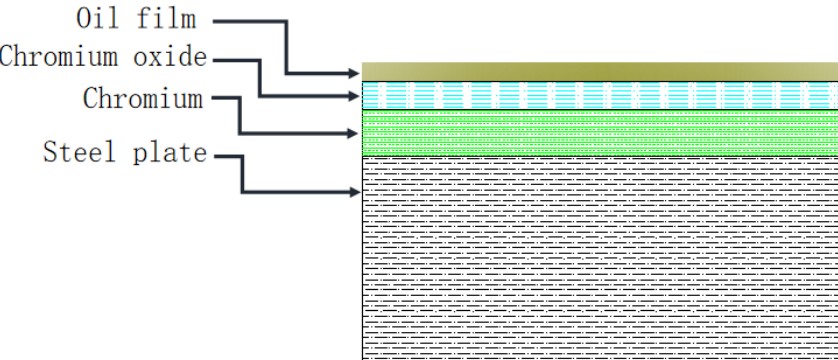

**Figure 5.** The composition structure of the TFS.

Inevitably, there is water vapor on the surface of chromium oxide, and hydroxide ions are generated. If the oil film or other pretreatment process is treated before generating the coating plate, it is possible to remove the hydroxide ions from the surface of the TFS, leaving only chromium oxide.

We analyzed the adsorption force of ester-based functional groups with nine crystalline surfaces of chromium oxide by simulation and selected the lattice index of chromium oxide with the greatest adsorption force. Based on this, a certain number of hydroxide ions was added, and then the adsorption force magnitude of chromium oxide with ester-based functional groups with the addition of hydroxide ions was analyzed.

The TFS surface inevitably has water vapor with the generation of hydroxide ions. If there is a heating process in the process of removing the oil film, it is possible to remove the hydroxide ions from the TFS surface.

We chose polyurethane coatings composed of polyester polyols, and the ester groups were selected for molecular dynamics analysis with $Cr_2O_3$ with several $Cr_2O_3$ surfaces [15,16]. The following surfaces were selected: $Cr_2O_3$ (100), $Cr_2O_3$ (110), $Cr_2O_3$ (111), $Cr_2O_3$ (012),

$Cr_2O_3$ (120), $Cr_2O_3$ (210), $Cr_2O_3$ (211), $Cr_2O_3$ (121)), and $Cr_2O_3$ (112). The model structures are shown in Figure 6, where the topmost layer representing the white marking line is the lattice surface. The distribution of oxygen and chromium atoms varies from lattice to lattice.

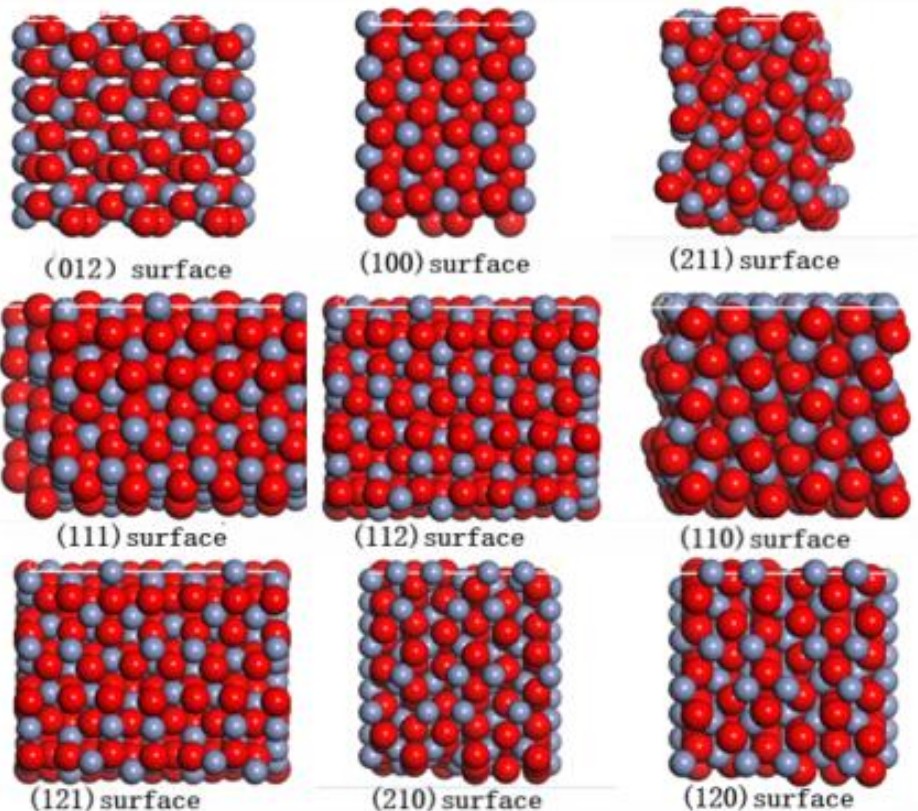

**Figure 6.** Atomic models of the selected $Cr_2O_3$ surfaces. (The gray balls are chromium atoms and the red balls are oxygen atoms).

The analysis process is shown in Figure 2. $Cr_2O_3$ is available in the software database, and the introduced chromium oxide cell is cut with nine Miller index crystal faces, and the thickness of the crystal face is assigned to form a new cell; new ester-based molecules are created in the software, and amorphous cell optimization [17] is performed. New ester-based molecules were created in the software and optimized for amorphous cells, and COMPASS III was chosen for the force field. There are many kinds of molecular force fields. Some molecular force fields are only used in a specific class or a number of classes of molecules. The advantage is their high accuracy, and the disadvantage is their lack of appropriate force field parameters, poor forecasting ability, and not being able to meet the requirements of the practical application. The COMPASS III force field is a force field that can be used to accurately simulate and predict the structure of a single molecule or condensed matter, conformation, vibrational frequency, and thermodynamic properties of the ab initio calculations in a wide range. The objects that can be studied by the COMPASS III force field include the most basic organic small molecules, inorganic small molecules, polymers, etc. It can also simulate and calculate many new types of materials formed by metal ions, metals, metal oxides, etc. The COMPASS III force field is suitable for the study of microscopic interfacial layers of metal oxides with organic molecules and polymers used in this study.

There can be four choices of coefficient synthesis for kinetic analysis, including NVE, NVT, NPH, and NPT. We chose NPH coefficient synthesis due to the need for pressure parameters. The equation of the pressure–volume relationship in the kinetic analysis is given by M. P. Allen [18] and others. The system synthesis belongs to the category

of statistical mechanics. The equation for calculating the temperature of the system in statistical mechanics is:

$$\langle P_k \frac{\partial H}{\partial p_k} \rangle = kT \tag{7}$$

The formula of the pressure is:

$$\langle q_k \frac{\partial H}{\partial q_k} \rangle = kT \tag{8}$$

The instantaneous pressure function derived from the pressure and temperature formula is:

$$P^* = \rho kt + \frac{W}{V} \tag{9}$$

$P^*$ is pressure; $\rho$ is density; and $V$ is volume.

This formula reveals the relationship between pressure, volume, and density. Constant pressure increases and volume decreases, constant pressure increases density and volume decreases. The interfacial layer formed the chromium oxide cell and ester-based molecule after the optimization of the amorphous cell, which is shown in Figure 7. The upper layer is the ester-based molecule and the lower layer is the chromium oxide cell, and a vacuum layer is established between them. According to the motion of the roll coating process, all chromium oxide atoms are fixed, and the ester-based molecules are not constrained.

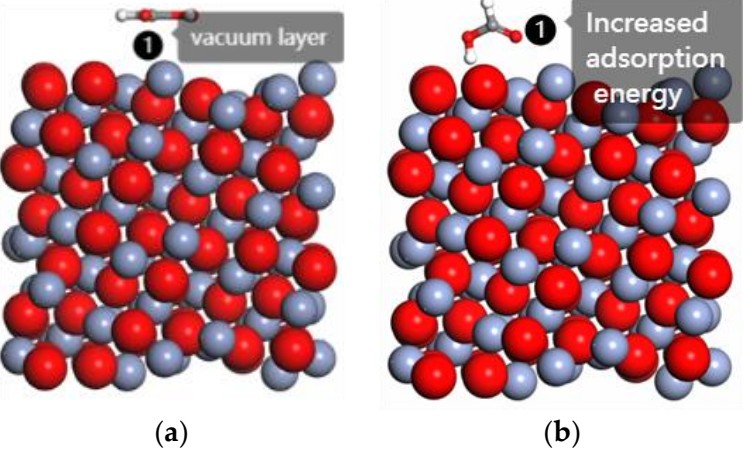

**(a)**            **(b)**

**Figure 7.** The TFS coating interfacial layer before optimization (**a**) and after optimization (**b**).

The interfacial layer was optimized by Forcite geometry optimization, and COMPASS III was chosen for the force field before and after the optimization of the interfacial layer, as shown in Figure 7. The method of using a force field to simulate molecular properties is the molecular mechanics method, which is performed calculating the potential energy of various possible conformations of molecules. The conformation with the lowest molecular potential energy is obtained, which is the most stable conformation. This process is the optimization process and is called energy minimization [19,20]. After optimization, the ester-based molecule undergoes a significant positional change and morphological change. The potential energy of the interfacial layer obtained at this time is the lowest and the structure is the most stable. We can consider this as the interfacial layer state that is closest to the actual situation.

The interfacial layer is optimized for molecular dynamics on the basis of geometric optimization, and the Forcite dynamics method is used to select the NVE system synthesis with 298 k temperature and COMPASS III for the force field [20–25]. During the dynamics simulation, the ester-based molecules undergo various displacement and attitude adjustment changes, and the energy change process is shown in Figure 3. As time continues, the total energy remains constant, the potential energy and non-bond energy decrease, and the kinetic energy increases. The interfacial layer energy is converted between kinetic

energy and potential energy. The ester group evolves to a dynamic equilibrium state after the influence of temperature and pressure, which prepares the final energy calculation. The next step is to calculate the total energy of the interfacial layer $E_{jmc}$, the energy of chromium oxide atoms $E_{yhg}$, and the energy of ester group molecules $E_{zhiji}$, and after the same process, nine crystal surfaces are calculated by adsorption energy $E_{xfn}$. The equation for the adsorption energy is derived from Equation (6) as follows.

$$E_{xfn} = E_{jmc} - \left( E_{yhg} + E_{zhiji} \right) \tag{10}$$

Using the simulation analysis process, we obtained the adsorption energy of the nine kinds of crystal surface, whose ranking chart is shown in Figure 8. The negative sign of the adsorption energy represents the adsorption force as gravitational force. The value of the adsorption energy of the (110) surface is the largest, and the adsorption force is also the largest. The (110) surface structure is shown in Figure 6. Chromium atoms are concentrated on the surface, and it is presumed that the more chromium atoms contained in the interface layer, the greater the adsorption energy. The specific value is shown in Table 1. There may be many kinds of chromium oxide crystal surfaces in actual situations, and they can also be synthesized artificially. In the synthesis of the crystal surface with the largest adsorption force, the interface layer can achieve a better adsorption effect, which has great practical significance for the roll coating process.

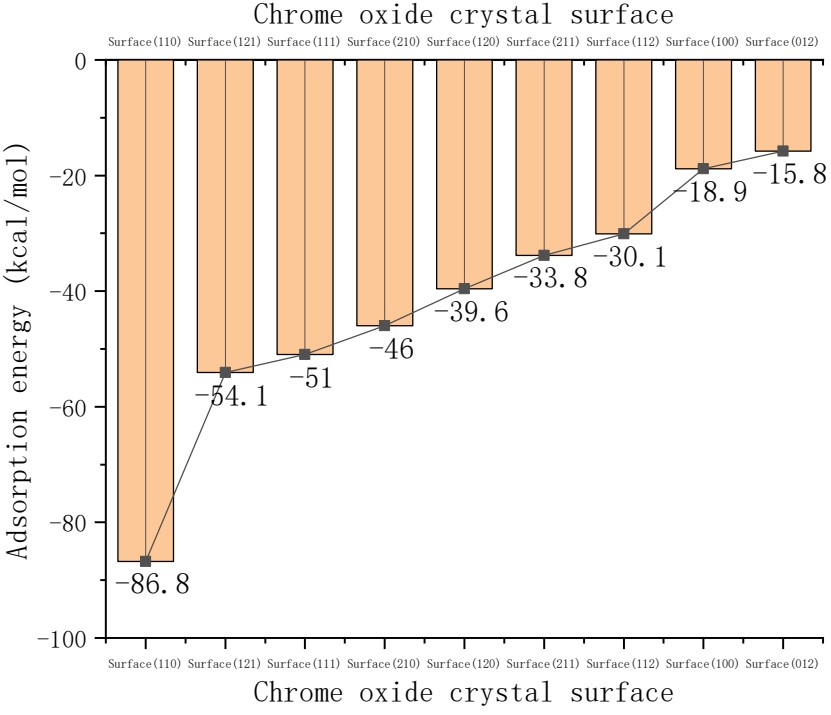

**Figure 8.** Adsorption energy corresponding to the nine crystal surfaces of chromium oxide.

In the future, we may study the chrome oxide crystal surface or find many kinds using the above ideas for adsorption energy sorting to find the best crystal surface index, which is also a direction of chromium plating plate process research.

**Table 1.** The adsorption energy of the ester groups corresponding to the nine crystal surfaces of chromium oxide.

| Crystal Surface | Total Energy $E_{jmc}$ | Chromium Oxide Energy $E_{yhg}$ | Ester Group Molecular Energy $E_{zhiji}$ | Adsorption Energy $E_{xfn}$ |
|---|---|---|---|---|
| $Cr_2O_3$ (100) | −59,232.10766 | −21.211918 | −59,192.03275 | −18.862986 |
| $Cr_2O_3$ (110) | −30,247.47664 | −12.539503 | −30,148.17373 | −86.763401 |
| $Cr_2O_3$ (111) | −98,680.97528 | −18.486182 | −98,611.51571 | −50.973385 |
| $Cr_2O_3$ (012) | −27,462.46695 | −23.911311 | −27,422.79271 | −15.762935 |
| $Cr_2O_3$ (120) | −158,228.6778 | −18.442777 | −158,170.6688 | −39.566181 |
| $Cr_2O_3$ (210) | −179,539.373 | −17.068746 | −179,476.3186 | −45.98569 |
| $Cr_2O_3$ (211) | −60,301.83219 | −17.26573 | −60,250.74186 | −33.824593 |
| $Cr_2O_3$ (121) | −119,487.8889 | −16.411669 | −119,417.4177 | −54.059536 |
| $Cr_2O_3$ (112) | −119,467.2344 | −19.72705 | −119,417.4177 | −30.089643 |

### 3.2. The Effect of Hydroxide Ions on Adsorption

The chromium plating plate surface may have hydroxide ions. We chose the (110) surface of chromium oxide, added a certain amount of hydroxide ions on its surface, as shown in Figure 9, and calculated its adsorption energy. The result was −123,749.929 kcal/mol. The results show that the surface contains hydroxide ions (110) on the surface of chromium oxide, in which adsorption force is a little higher. So, we tried not to destroy it before the roll coating process of hydroxide ions, which helps to improve the adsorption of the coating. The chromium plating plate processing process tries to retain the hydroxide ion as much as possible, which also has a positive effect on the subsequent roll coating process. This provides an important reference for the manufacturing process of the TFS.

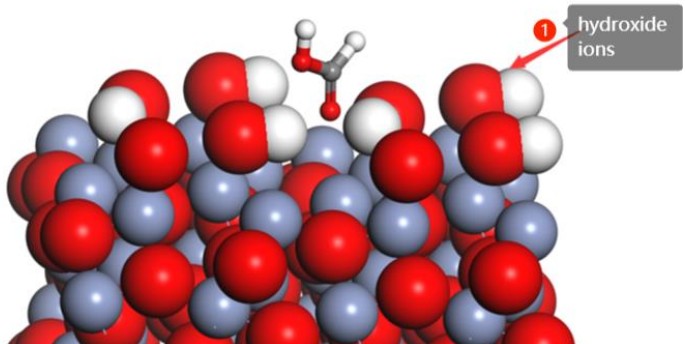

**Figure 9.** Addition of hydroxide ions to the chromium oxide surface of the (110) surface.

### 3.3. Interfacial Layer Adsorption Type

The molecular dynamics analysis allows us to determine whether the type of adsorption in the interfacial layer is chemisorption or physisorption. Physisorption means that the force between the molecules of the fluid is being adsorbed, and the molecule of the solid surface is intermolecular attraction, which is known as the Van der Waals force. Therefore, physisorption is also known as Van der Waals adsorption, where the Van der Waals surfaces of the atomic surfaces will be tangent or separated in the case of physisorption, and the Van der Waals surfaces of the atoms will intersect in the case of chemisorption. As shown in Figure 10, the middle blue surface is the Van der Waals surface and the Van der Waals surface issues of ester-based molecules and chromium oxide intersect, which has surpassed the molecular force of physisorption; so, the ester-based molecules of chromium oxide behave as chemisorption [26–30]. And the adsorption energy magnitude in Figure 8 also far exceeds the range of physisorption, which also proves that the adsorption type of the interfacial layer formed by the coating and chromium oxide surface is mainly chemisorption.

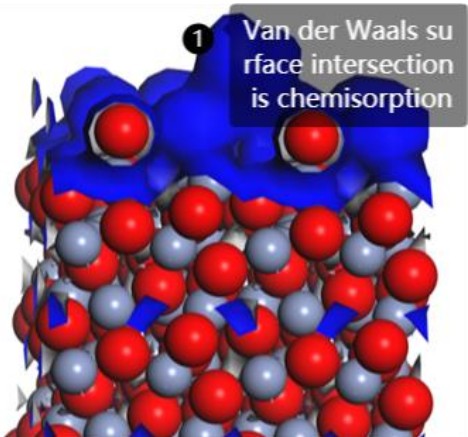

**Figure 10.** Van der Waals force boundaries for chromium oxide on the (110) surface.

*3.4. Analysis of the Adsorption of Polymer Functional Groups in Coatings*

The microscopic molecular structure of the coatings determines the adsorption performance of the coatings. There are many functional groups in the coatings, and by analyzing the functional groups, different functional groups can be analyzed to show different adsorption, so the ranking of the influence of functional groups on the size of adsorption can be obtained by analyzing the adsorption of common functional groups and the interfacial layer of chromium oxide on the (110) surface. In this way, it is possible to determine the influence factor of coatings on the size of interfacial layer adsorption based on the type and number of functional groups contained in the coatings. The common functional groups are amino, methylene, cyano, ether, and ester groups, and the adsorption energy data of each functional group are obtained by a molecular dynamics analysis of the process shown in Figure 2. The adsorption energy data are shown in Figure 11. The molecular dynamics analysis of only these five common functional groups was performed in this study, and the analysis and calculation of adsorption energy of more functional groups, as well as experimental verification, can be performed in a subsequent study.

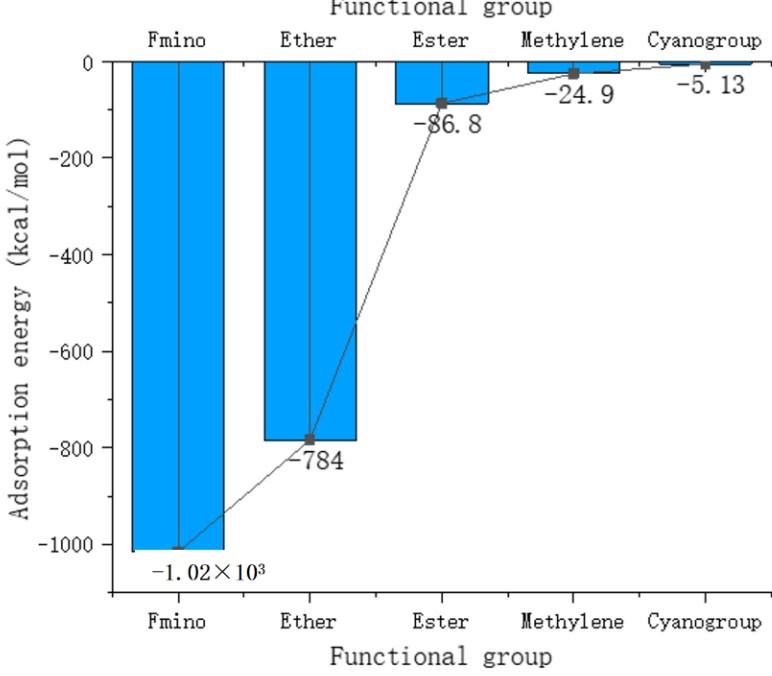

**Figure 11.** Adsorption energy corresponding to common functional groups.

The composition of the coatings and the percentage of the number of functional groups can be provided by the coatings manufacturer or detected themselves. Many institutions have prepared a library of infrared spectroscopy standard spectra, which only needs to be compared to determine the composition of the coatings sample to be tested. By sorting, the adsorption energy of different coatings to chromium oxide can be sorted, and finally, the coatings sample to be tested can be compared with the benchmark coatings to find out whether the adsorption force increases or decreases, and then the roll coating process parameters, namely pressure P1, and the size of P1 are determined by the center distance (a2) between the cover roller and the pressure-bearing roller, and guidance is given to adjust a2 to increase or decrease.

We used molecular dynamics simulation for further study. The difference between polyester polyol and polyether polyol is the ester group and ether group. By comparing the adsorption energy of the ester group and the ether group, the adsorption force of the ester group is obviously higher than the ether group, so it is deduced that the adhesion force of polyester polyol in the interfacial layer is greater than polyether polyol in the interfacial layer, and it is further deduced that the adsorption force of polyester type polyurethane is greater than polyether type polyurethane.

Let us simulate the analysis using the method in Figure 2. The adsorption energy of polyester polyol in the interfacial layer is −361.67957 kcal/mol, the adsorption energy of polyether polyol in the interfacial layer is −249.393985 kcal/mol, and the negative sign is the gravitational force, so the adsorption energy of polyester polyol in the interfacial layer is greater than the adsorption energy of polyether polyol in the interfacial layer. The adsorption energy of polyester polyurethane is −1451.6578 kcal/mol, the adsorption energy of polyether polyurethane is −573.8439 kcal/mol, and the negative sign is the gravitational force, so the adsorption energy of polyester polyurethane in the interfacial layer is greater than the adsorption energy of polyether polyurethane in the interfacial layer.

Taking polyurethane as an example to analyze the adsorption force, Wu Guohua [31] performed a detailed analysis of its adsorption force by an experimental method, and its adsorption force size was determined by the drawing circle method. The adsorption force of polyester-type polyurethane was greater than polyether-type polyurethane. Li Shaoxiong et al. [32] theoretically analyzed that the adsorption force of polyester-type polyurethane is greater than polyether-type polyurethane because of the greater polarity of the ester group and the higher cohesion energy of the ester group (12.2 kJ/mol) compared to the ether group (4.2 kJ/mol), which leads to a greater adsorption force with the substrate compared tothe polyether-type polyurethane due to the polar effect of the ester bond. Both of them were used to prove that the adsorption force of polyester-type polyurethane is greater than polyether-type polyurethane by an experimental method and theoretical analysis, respectively.

*3.5. Experimental Verification*

The purpose of the experiment is to analyze the results and carry out experimental verification. At present, all the traditional methods cannot determine the actual adsorption size or adsorption energy size. This can only be performed by means of spacing to prove the grade of adsorption. In order to verify the adhesion of polyester polyol and polyether polyol, you can use the traditional conventional methods such as the scratch circle method, scratch grid method, pull apart method, and scratch X method for adhesion testing, but these methods are are based on the probability statistics of the phenomena, which are more random and, therefore, not accurate. In order to obtain more accurate experimental results, a more precise experimental scheme was designed, as shown in Figure 12, where the coating plate substrate is a chrome-plated plate, and both sides of the TFS are polished to increase the roughness and increase the adhesion.

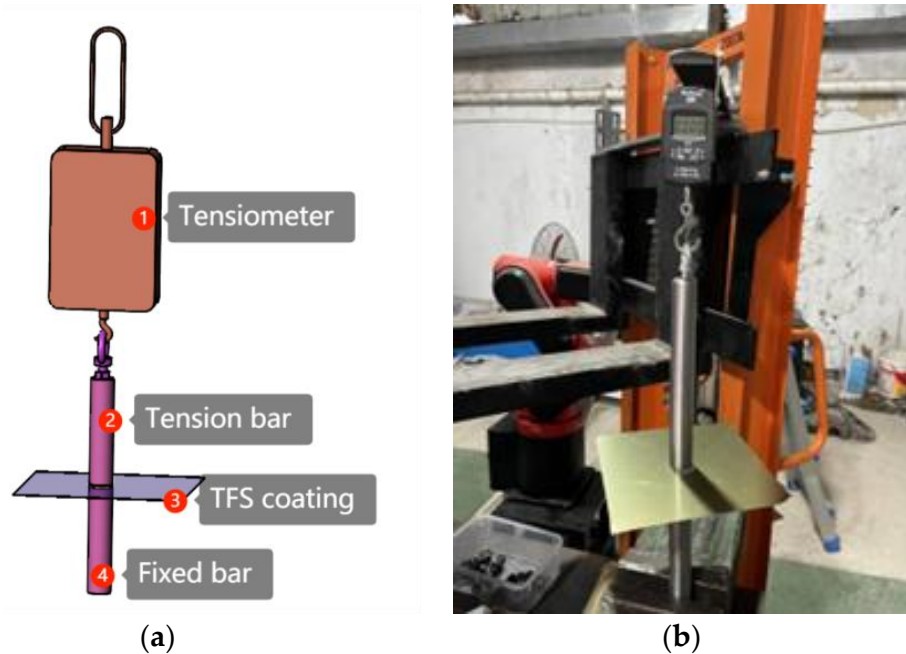

(**a**)                                           (**b**)

**Figure 12.** Experimental scheme for testing the magnitude of the adsorption force. (**a**) Experimental schematic, (**b**) experimental equipment.

The operation principle diagram of the roller coating machine is shown in Figure 13. The center distance between the screed roller and overmoulding roller is $a1$, the pressure between the screed roller and overmoulding roller is P1, the center distance between the overmoulding roller and bearing roller is $a2$, the pressure between the overmoulding roller and bearing roller is P2, the thickness of wet film on the TFS coating board is $h$, the thickness of the dry film is $t1$, and the percentage of solids of coating is β. The thickness of constant adhering coating on the screed roller is $b1$, the thickness of the coating between the screed roller and the thickness of the coating between the cover roller is $h1$, the thickness of the coating transferred by the cover roller is $b2$, the thickness of the coating adhering to the cover roller is $b3$, and the thickness of the coating between the cover roller and the TFS plate is $h2$, according to the relationship between the thickness of the rolling process:

$$h1 = b1 + b2 \tag{11}$$

$$h2 = b2 = h + b3 \tag{12}$$

$$t1 = h * β \tag{13}$$

$$h = h1 - b1 - b3 \tag{14}$$

Equation (14) shows that the wet film thickness h of the TFS board is determined with $h1$, $b1$, and $b3$. Assuming that $b1$ and $b3$ are constants in the roll coating process, the thickness of $h$ is determined by $a1$. $a1$ and $a2$ can be adjusted by the handwheel in the roll coating process. $a1$ and $a2$ affect physical parameters P1 and P2. So, $h$ is determined by $p1$. According to Equation (13), $t1$ is determined by $a1$. According to the current roll coating process, the a1 increase and $a2$ constant can increase the coating thickness $t1$ and increase the denseness. If the process requirement is to increase the coating thickness and the denseness is unchanged, then it is necessary to increase $a1$ and decrease $a2$. The adjustment of $a1$ and $a2$ affects the coating thickness and denseness, which essentially affects the pressure $p1$ and $p2$. $p1$ affects the dry film thickness $t1$, and the size of $p2$ then affects the adsorption

force between the coating and the substrate. Therefore, the subsequent experiments mainly verify the relationship between $p1$ and $p2$ and the influence of adsorption and compactness.

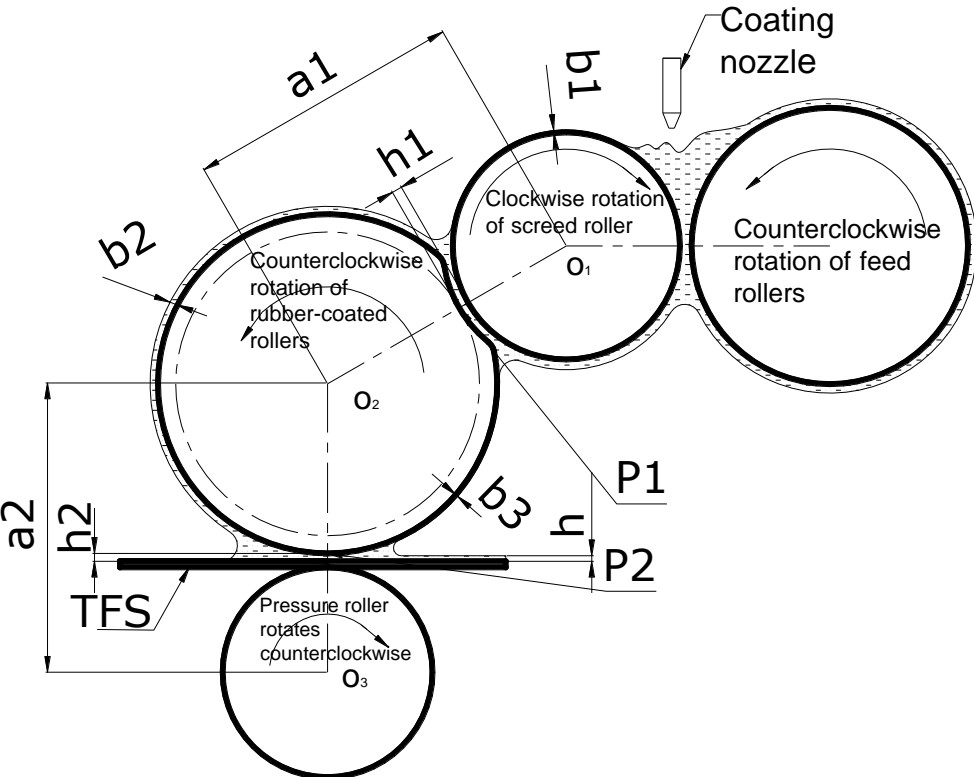

**Figure 13.** Roll coating machine running principle.

To summarize, the adsorption and denseness of the coating are mainly influenced by the coating composition, chromium oxide crystalline surface, and pressure. The chromium oxide crystalline surface can be measured, and the coating composition is changed by molecular simulation analysis to determine the required compactness and adsorption energy and thus the pressure $p2$. The coating thickness is regulated by $p1$. The experimental scheme is shown in Figure 12. According to the principle of Newtonian mechanics, both sides of the paint plate have equal tension before disengagement, and the first side to be disengaged has a small adsorption force.

The experimental supplies are shown in Figure 14, with polyester polyol on the left, polyether polyol in the middle, and the TFS on the right. These three materials are the consumables needed for the experiment. The tools required are a hydraulic forklift, a fixed rod, and a tension rod. The experimental procedure is shown in Figure 13, where the hydraulic forklift is raised to pull away the experimental coating. The physical properties of polyether polyol and polyester polyol are shown in Table 2. These two raw materials are liquid at room temperature and are not toxic and suitable for the roll-on process.

**Table 2.** Physical properties of polyester polyols and polyether polyols.

| Name | Condition | Toxic or Not | Color | Coating Method |
|---|---|---|---|---|
| Polyether polyol | Liquid | No | Transparent | Roll coating |
| Polyester polyol | Liquid | No | White | Roll coating |

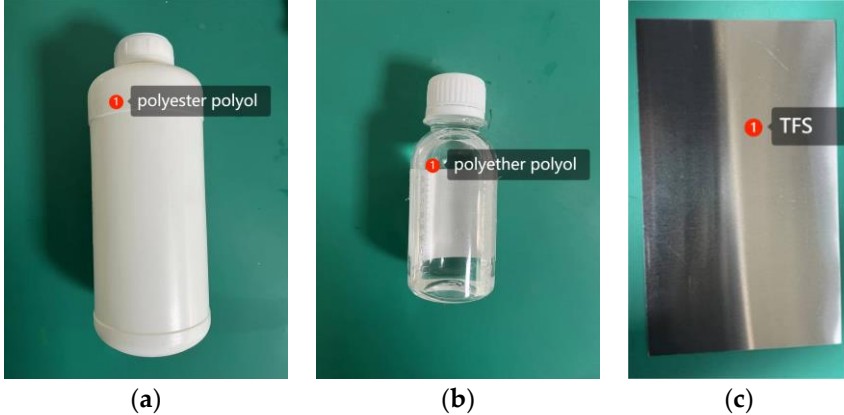

**Figure 14.** Experiment materials: (**a**) polyester polyol, (**b**) polyether polyol, (**c**) TFS.

Experiment 1: The front and back sides of the coating board were coated with polyester polyol and polyether polyol, the tension rod and the fixed rod were glued to both sides of the coating board by a strong adhesive, the ring of the tension scale was pulled up by a hydraulic forklift, and the adsorption force on the detached side was small. Record the reading w1 of the pulling force scale at this time; continue to apply strong adhesive on the detached side and re-detect the pulling force scale degree w2 when detached.

Experiment 2: The front and back sides of the coating board are coated with a polyester polyurethane coating and a polyether polyurethane coating, the pulling rod and the fixed rod are glued to both sides of the coating board by a strong adhesive, the ring of the pulling force meter is pulled up by a hydraulic forklift, and the adsorption force on the detached side is small. Record the reading w3 of the pulling force scale at this time; continue to apply strong adhesive on the detached side and re-detect the pulling force scale degree w4 when detached.

The results of the experiments were:
Experiment 1: polyether polyol side first off; w1 = 10.54 kg; w2 = 13.74 kg.
Experiment 2: polyester-type polyurethane coating comes offside first; w3 = 14.48 kg; w4 = 17.79 kg.

The simulation analysis, experimental results, and theoretical analysis results prove each other, and the conclusion is credible.

## 4. Discussion of the Compactness of the Interface Layer

### 4.1. A Study of the Morphology and Location of Polymers under Different Pressures

The compactness of metal packaging coatings is not the greater the better. In order to meet the processing requirements, a certain degree of flexibility and ductility is needed. Some occasions need to retain a certain gap to allow some contact between the metal ions so that the material inside can ensure the flavor. For example, a tin coatings plate in contact with the can solution can ensure the flavor of the can. This requires figuring out the microscopic behavior of the polymer at the interfacial layer under the influence of macroscopic temperature and pressure. The goal is to meet the compactness requirements only at lower pressures. This is the best state for the roll coating process.

The research object is a commonly used polyurethane coating. Polyurethanes are divided into polyether polyurethanes and polyester polyurethanes; the difference is whether the soft chain is composed of polyester polyol or polyether polyol. This molecular formula contains the main functional groups of polyester polyurethanes and is geometrically and energetically optimized to make it closest to the morphological state of the actual coating. The amorphous cell module, which provides a comprehensive set of tools to construct the three-dimensional periodic structure of the polymer system, was used to construct the molecule in a Monte Carlo fashion by minimizing the gaps between atoms while ensuring the torsional angle of any given force field for the conformation, which uses the COM-

PASS III force field and geometry optimization using the Forcite dynamics method. The purpose of using these optimization methods is to achieve a simplified molecular formula for polyester-type polyurethanes that is close to the state that is exhibited in the actual working conditions. After optimization, the morphology of the polyester polyurethane molecules under different pressures was analyzed, as shown in Figure 15. The polymer state is relatively loose, and the molecular morphology becomes more aggregated as the pressure increases.

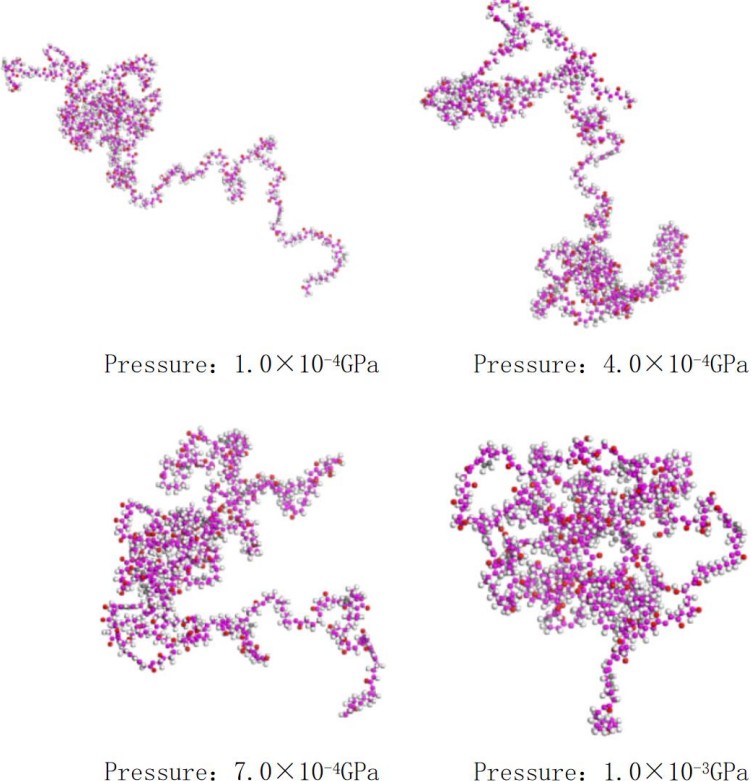

Pressure: $1.0 \times 10^{-4}$GPa     Pressure: $4.0 \times 10^{-4}$GPa

Pressure: $7.0 \times 10^{-4}$GPa     Pressure: $1.0 \times 10^{-3}$GPa

**Figure 15.** Polyester-type amino acids simulate the morphology of molecular formulae under different pressures.

The kinetic optimization of polyester-type polyurethane molecules was performed using NPT system synthesis [33], which allows the introduction of macroscopic temperature and pressure parameters. We used a temperature of 298 k and pressures of $1.0 \times 10^{-4}$ GPa, $4.0 \times 10^{-4}$ GPa, $7.0 \times 10^{-4}$ GPa, and $1.0 \times 10^{-3}$ GPa, respectively. As shown in Figure 16, polyester-type polyurethane molecules at different pressures have molecular densities with increasing pressures. The values of the density with increasing pressure are 1.15761924132913 g/cm$^3$, 1.163803755 g/cm$^3$, 1.17976811063181 g/cm$^3$, and 1.18946839571774 g/cm$^3$, respectively. This further indicates that the denseness of the coating increases with increasing pressure, which is also consistent with actual roll coating process experience.

The substrate still uses a chromium oxide plate, and the crystalline surface (110) chromium oxide cells form an interfacial layer with polyester-type amino acid molecules and the kinetic simulation uses NPT system synthesis and introduces macroscopic temperature and pressure parameters. We used a temperature of 298 k and pressures of $1.0 \times 10^{-4}$ GPa and $1.0 \times 10^{-3}$ GPa, respectively. As shown in Figure 17, at a temperature of 298 k and $1.0 \times 10^{-4}$ GPa of pressure, the polyester-type amino acid molecules are spaced from the crystalline surface (110) chromium oxide cells, and most of the atoms of the polyester-type amino acid molecules are concentrated in the upper part.

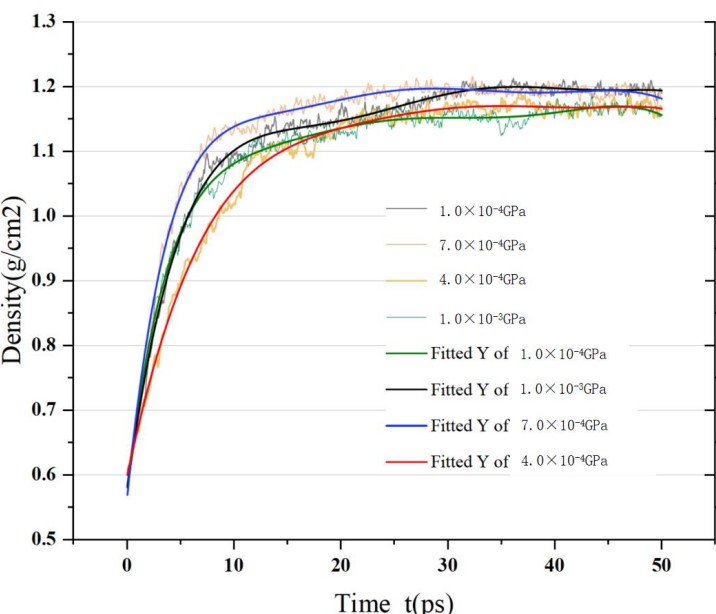

**Figure 16.** Curves of the density of polyester-type amino acids at different pressures with time.

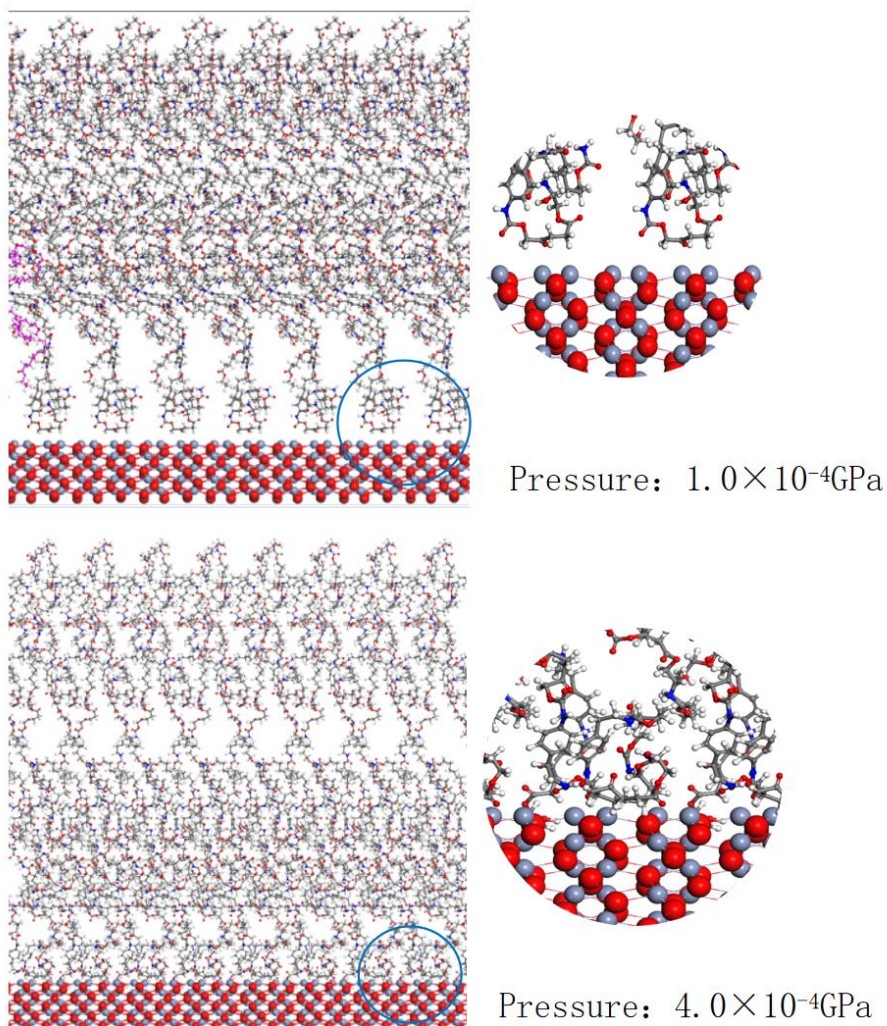

**Figure 17.** Interfacial layer formed by polyester-type amino acid molecules at the $Cr_2O_3$ (110) surface (**top**: pressure $1.0 \times 10^{-4}$ GPa; **bottom**: pressure $4.0 \times 10^{-4}$ GPa).

The compactness of the polymer is also expressed in its posture and morphology, and the change in its denseness is observed by looking at the morphology of its molecular formula in three dimensions, as shown in Figure 17, which is the simulated morphology of the polyester-type amino acid molecule in the interfacial layer at room temperature and standard atmospheric pressure. There is a certain gap in the interfacial layer, and most of the atoms and functional groups are distributed in the upper part, which seems to be a floating unadsorbed state. With the increase in pressure, the simulated morphology of ester-type amino acid molecules in the interfacial layer tends to move toward the surface of chromium oxide, and the denseness increases and the gap becomes smaller, which is also a characterization of the increase in adsorption. Figures 17 and 18 show that the position of the polyester-type amino acid molecule moves toward the chromium oxide cell with increasing pressure, and the morphology and attitude also change. Most of its atoms move toward the chromium oxide cell, which macroscopically shows an increase in density and compactness.

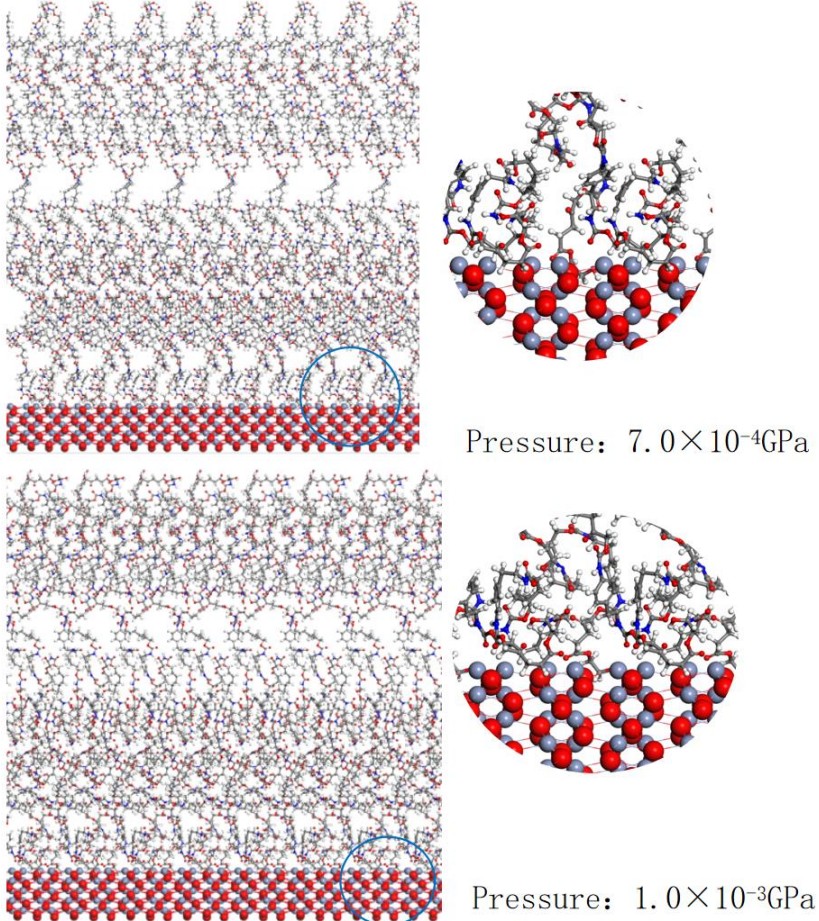

**Figure 18.** Interfacial layer formed by polyester-type amino acid molecules at the $Cr_2O_3$ (110) surface (**top**: pressure $7.0 \times 10^{-4}$ GPa; **bottom**: pressure $1.0 \times 10^{-4}$ GPa).

## 4.2. Experimental Verification

The compactness experiment is mainly characterized by the density of the coating. First of all, we need to determine the pressure value between the cover roller and the TFS coating board. We use a thin film sensor with a digital display-led data acquisition card, as shown in Figure 19. The measured weight values are converted into pressure values of $1.0 \times 10^{-4}$ GPa, $4.0 \times 10^{-4}$ GPa, $7.0 \times 10^{-4}$ GPa and $1.0 \times 10^{-3}$ Gpa. The pressure test method is shown in Figure 19.

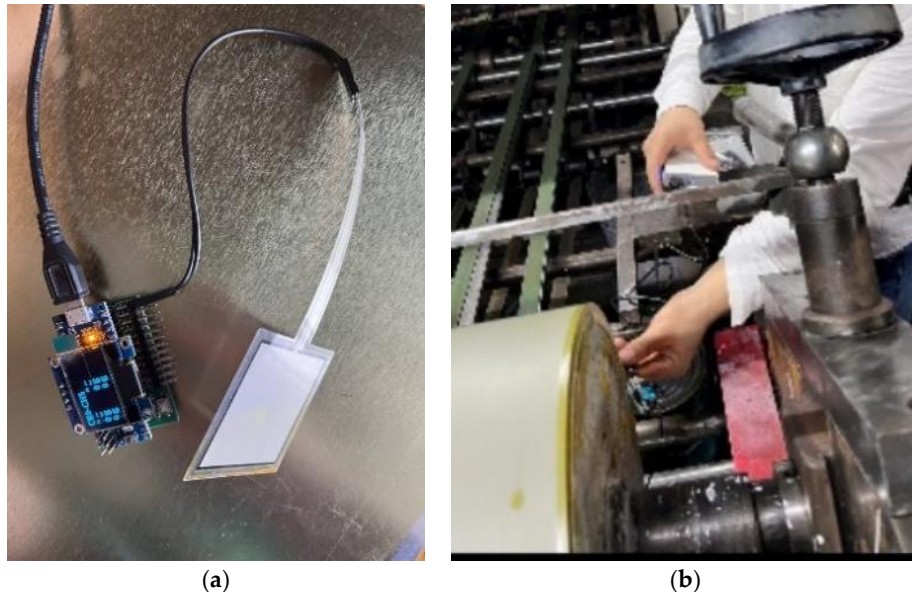

**Figure 19.** Thin film pressure sensors detect pressure. (**a**) Thin film pressure sensor, (**b**) thin film sensor test methods.

SEM observed the morphology of the interfacial layer under different pressures, and the applied pressure gradually increased in the order of (a) to (d), as shown in Figure 20. It can be seen from the sample of the transition layer that the transition layer will gradually become thinner because the adsorption force and compactness of the interfacial layer will increase with the increase in pressure.

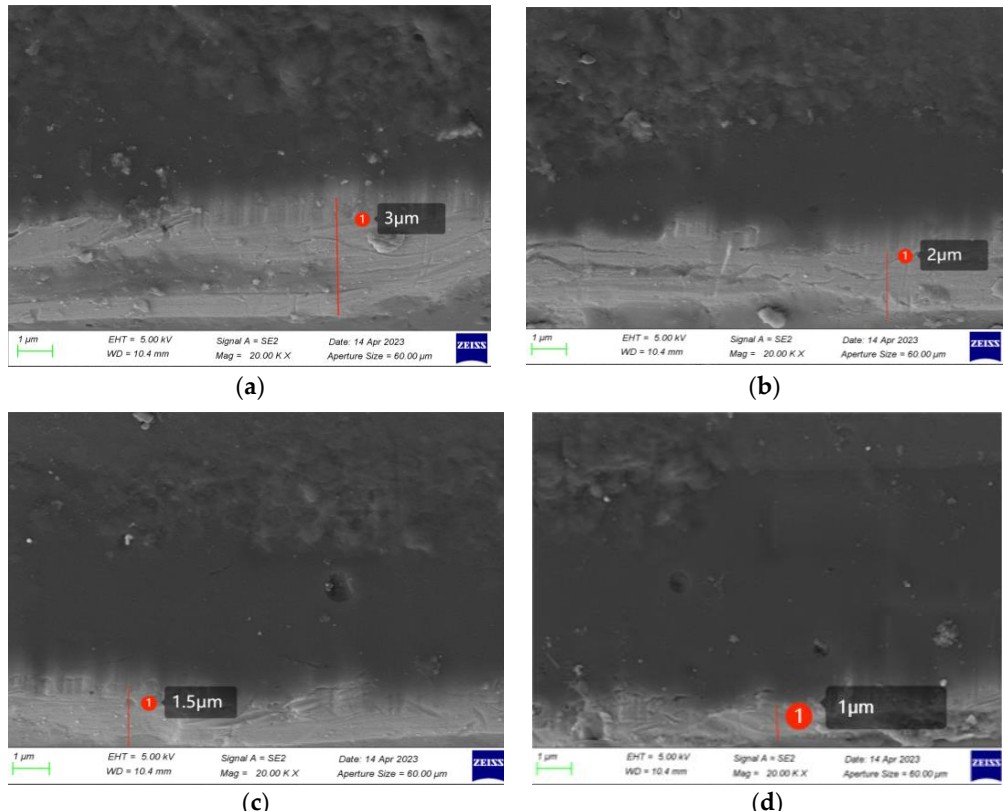

**Figure 20.** The morphology of the interfacial layer was observed by SEM under different pressures. (**a**) Transition layer thickness ($1.0 \times 10^{-4}$ GPa); (**b**) Transition layer thickness ($4.0 \times 10^{-4}$ GPa); (**c**) Transition layer thickness ($7.0 \times 10^{-4}$ GPa); (**d**) Transition layer thickness ($1.0 \times 10^{-3}$ GPa).

As shown in Figure 21a, the pressure detection value is 18,940 g, which is converted into a pressure of $0.966 \times 10^{-3}$ GPa, and the thickness of the transition layer is about 1 μm, as observed by electron microscope. The density of the coating layer is calculated by weighing on a high-precision balance in Figure 21b and the average value of multiple sets of data is $1.18348 \text{ g/cm}^3$ compared with the simulated data of $1.18272 \text{ g/cm}^3$, as shown in Figure 16. After experimental verification, the value of the macroscopic film pressure sensor is 18,940 g when the coating meets the requirements of adsorption and compactness.

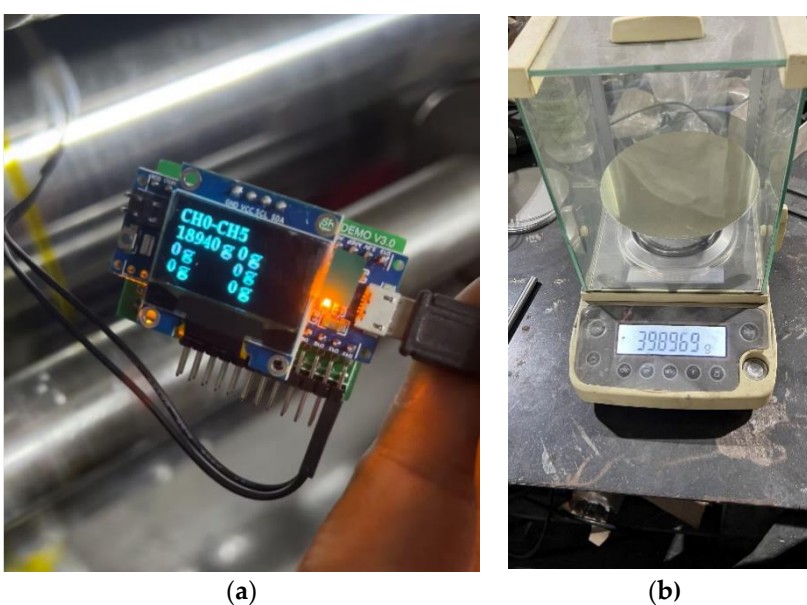

| (a) | (b) |

**Figure 21.** Pressure and weight testing. (**a**) Thin film pressure sensor values; (**b**) High-precision balance scales.

Based on the summary of the test results, the adsorption force can be accurately quantified using the above test method, and the compactness of the interfacial layer can be quantified by density. The roll coating process to meet the adsorption and densification requirements can be adjusted with precise values by a thin film pressure sensor.

## 5. Conclusions

The adsorption and compactness of the interfacial layer of the TFS-coated plate can be ranked qualitatively by molecular simulations. The simulation analysis shows that the maximum adsorption force of chromium oxide is on the crystalline surface (110), and the adsorption force becomes larger with the addition of hydroxide ions on the chromium oxide crystalline surface. The polymer in the coating is composed of various functional groups and the adsorption force is related to the type of functional group. The adsorption of the polyester group is greater than the polyether group, and the adsorption of polyester polyol is greater than polyether polyol; the adsorption of the polyester-type polyurethane coating is greater than the polyether-type polyurethane coating. It is deduced that the adsorption of the coatings can be ranked qualitatively according to the type and number of functional groups; the adsorption of the interfacial layer is mainly expressed as chemisorption. The denseness of the interfacial layer increases with the increase in pressure; the density of the coating also increases with the increase in pressure. The increase in compactness in the microscopic interfacial layer is manifested by the aggregation of polymer molecules in the interfacial layer.

After experimental verification, the pressure and density of the interfacial layer of the TFS coating plate can correspond to the macroscopic data. The adsorption and compactness to meet the process requirements have a corresponding quantitative parameter, which is the value displayed by the film sensor, and for the coatings used in this experiment, only

the film pressure needs to be adjusted to 1,8940 g to meet the process requirements. The experimental, theoretical, and simulation analyses mutually validate a feasible scheme for analyzing the adsorption and compactness of the coating plates, which is the theoretical result of this study. The process parameters affecting adsorption and compactness can be adjusted quantitatively, which is the practical result of this study. The results of this study only investigated the microscopic interfacial layer of compactness and adsorption of TFS surface coatings, and the experiments were from the roller coating process, which has some limitations, but the research method can be extended to other processes (brushing, spraying, etc.) to study the adsorption mechanism and compactness analysis of coatings on the surface of coatings and substrates.

**Author Contributions:** Conceptualization, Y.X. and T.Z.; methodology, Y.X.; software, Y.X.; validation, Y.X., T.Z. and K.L.; formal analysis, Y.X.; investigation, K.L.; resources, K.L.; data curation, Y.X.; writing—original draft preparation, Y.X.; writing—review and editing, Y.X.; visualization, Y.X.; supervision, K.L.; project administration, K.L.; funding acquisition, T.Z. All authors have read and agreed to the published version of the manuscript.

**Funding:** This research received no external funding.

**Institutional Review Board Statement:** Not applicable.

**Informed Consent Statement:** Not applicable.

**Data Availability Statement:** All data in this work are available upon request by contacting the corresponding author.

**Acknowledgments:** The authors are grateful for the support from Suzhou Panjin New Material Co., LTD, Jiangsu.

**Conflicts of Interest:** The authors declare no conflict of interest.

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
