# Peer review of "A Study on the Adsorption Mechanism and Compactness of the TFS Coating Interfacial Layer"

_coatings, doi:10.3390/coatings13071290_

Round 1
Reviewer 1 Report
The paper represents an interesting interdisciplinary study on coatings applicable in sensoric devices. Both theoretical and engineering approaches are convincingly exploited. The paper could be published in the present form.
Author Response
Thank you very much for your recognition!
Reviewer 2 Report
The study of the production and use of tin free steel (TFS) has been going on for quite a long time (especially regarding the production of food packaging). It should be noted here that tin free steel or electrolytically chromium, chromium oxide coated steel is similar to tin plate except, in most cases, non-involvement of flow melting and chemical passivation during its production. The use of TFS is less as compared to tin plate and mainly utilized for food can ends, crown caps, and vacuum closures for glass containers. Removal of coatings as a prerequisite for welding of TFS hinders its extensive usage for single use containers and recyclability. Nevertheless, its low cost over tinplate makes it the best choice for drums used in bulk storing and transportation of finished products. Therefore, i think that the research topic is relevant and interesting.
In the present study, the molecular simulation model of interfacial layer interaction of TFS coating plate was established by using molecular mechanics simulation, Monte Carlo simulation and molecular dynamics simulation, and the influential rules of chromium oxide crystalline structure, coating functional group type and coating pressure on the adsorption and compactness of interfacial layer were analyzed and verified by experiments. I believe that the research results have scientific novelty and practical value.
There are several comments and recommendations to the text of the manuscript.
1. Please check the traditional names for research manuscript sections are correct. In particular, section 2 should be entitled "Materials and Methods". It should be followed by the sections "Results", "Discussion", "Conclusions" (You can see more details about the requirements for the structure of a research article in the mdpi publishing house here: https://www.mdpi.com/journal/coatings/instructions#preparation).
2. At the end of the "1 Introduction" section, you outlined the motivation for your research. After that, it is necessary to clearly and unambiguously formulate the purpose of the study.
3. Over the past 5 years, quite a lot of research has been published regarding the production, practical use and optimization of the properties of tin free steel. In the "1 Introduction" section, you need to cite the main new research that in one way or another relates to the topic of your manuscript. Overview of the field where a similar concept is being developed is important to claim novelty and help readers what they are looking for.
4. The force field is called the soul of molecular mechanics and molecular dynamics, and naturally that the choice of force field directly determines the accuracy of simulation results. It is necessary to justify in the text why you chose COMPASS III for your study.
5. On Figure 13 "Roll coating machine running principle" there are inscriptions in Chinese, they must be replaced or duplicated in English.
6. Recommendation for improvement of conclusions. In your conclusions, you presented the results well. Nevertheless the findings and their implications should be discussed in the broadest context possible, and also indicate the applied value of the research. It is also necessary to indicate the limitations of the results or methods obtained and outline the direction of further research.

Author Response
Dear Editor:
Thank you very much for your valuable inputs I have made the changes in my paper, screenshots below.
- Please check the traditional names for research manuscript sections are correct. In particular, section 2 should be entitled "Materials and Methods". It should be followed by the sections "Results", "Discussion", "Conclusions" (You can see more details about the requirements for the structure of a research article in the mdpi publishing house here: https://www.mdpi.com/journal/coatings/instructions#preparation).
I have made the requested formatting terminology changes.
- At the end of the "1 Introduction" section, you outlined the motivation for your research. After that, it is necessary to clearly and unambiguously formulate the purpose of the study.
I added the purpose of the study here.
- Over the past 5 years, quite a lot of research has been published regarding the production, practical use and optimization of the properties of tin free steel. In the "1 Introduction" section, you need to cite the main new research that in one way or another relates to the topic of your manuscript. Overview of the field where a similar concept is being developed is important to claim novelty and help readers what they are looking for.
I added references and the purpose of their study and made a thesis statement.
- The force field is called the soul of molecular mechanics and molecular dynamics, and naturally that the choice of force field directly determines the accuracy of simulation results. It is necessary to justify in the text why you chose COMPASS III for your study.
I added this part.
- On Figure 13 "Roll coating machine running principle" there are inscriptions in Chinese, they must be replaced or duplicated in English.
I've corrected it.
- Recommendation for improvement of conclusions. In your conclusions, you presented the results well. Nevertheless the findings and their implications should be discussed in the broadest context possible, and also indicate the applied value of the research. It is also necessary to indicate the limitations of the results or methods obtained and outline the direction of further research.
Thank you very much for this comment, I have added the narrative accordingly!

Reviewer 3 Report
This is a practical work with modeling elements. It concerns the study of adsorption mechanism and compactness of TFS coating interfacial layer. I think that the experimental plan was properly drawn up and carried out, and the results obtained are of great practical importance and can be used by industry.
There are several editorial errors that should be removed:
1. Figure captions are in bold or non-bold font. A period is missing after the figure number. This should be corrected and standardized.
3. Figure 12 - the photo of the test stand is too small - not much can be read from it.
2. The descriptions of the elements placed in Figure 13 are not in English. I am not able to read them.
Author Response
Dear Editor:
Thank you very much for your valuable inputs I have made the changes in my paper, screenshots below.
- Figure captions are in bold or non-bold font. A period is missing after the figure number. This should be corrected and standardized.
I've standardized the headings to be bold
3. Figure 12 - the photo of the test stand is too small - not much can be read from it.
I've made changes and enlarged the picture
2. The descriptions of the elements placed in Figure 13 are not in English. I am not able to read them.
I've translated it into English.
